
# A newly reconciled data set for identifying sea level rise and variability in Dublin Bay

Amin Shoari Nejad[1], Andrew C Parnell[1], Alice Greene[2], Peter Thorne[2], Brian P Kelleher[3], Robert J.N. Devoy[4], and Gerard McCarthy[2]

[1]Hamilton Institute, Insight Centre for Data Analytics, Maynooth University, Kildare, Ireland
[2]ICARUS, Department of Geography, Maynooth University, Maynooth, Ireland
[3]School of Chemical Sciences, Dublin City University, Dublin 9, Ireland
[4]Department of Geography, University College Cork, Cork, Ireland

**Correspondence:** Amin Shoari Nejad (amin.shoarinejad.2020@mumail.ie)

**Abstract.** We provide an updated sea level dataset for Dublin for the period 1938 to 2016 at yearly resolution. Using a newly collated sea level record for Dublin Port, as well as two nearby tide gauges at Arklow and Howth Harbour, we perform data quality checks and calibration of the Dublin Port record by adjusting the biased high water level measurements that affect the overall calculation of mean sea level (MSL). To correct these MSL values, we use a novel Bayesian linear regression

that includes the Mean Low Water values as a predictor in the model. We validate the re-created MSL dataset and show its consistency with other nearby tide gauge datasets. Using our new corrected dataset, we estimate a rate of 1.08 mm/yr sea level rise at Dublin Port during 1953-2016 (95% CI from 0.62 to 1.55 mm/yr), and a rate of 6.48 mm/yr during 1997-2016 (95% CI 4.22 to 8.80 mm/yr). Overall sea level rise is in line with expected trends but large multidecadal variability has led to higher rates of rise in recent years.

## 1   Introduction

Global mean sea level is rising due to anthropogenic climate change (Devoy, 2015; Masson-Delmotte et al., 2021). Understanding regional sea level trends is crucial for local and regional adaptation and the development of effective climate action plans. In Ireland, Dublin is the largest city with a population of approximately 1.42 million (CSO, 2019) and is situated at the mouth of the river Liffey on the Irish Sea coast. Dublin also has the Republic of Ireland's longest tide gauge record data from

Dublin Port (also called Dublin North Wall) publicly available from 1938 onwards. Understanding changes in mean sea level in Dublin is key for the protection of Ireland's largest city and, from a national perspective, in understanding long-term sea level rise (SLR) in Ireland (DCC, 2005).

Sea level around Ireland rose rapidly after the last glacial maximum 20,000 years ago, cutting Ireland off as an island 16,000 years ago (Edwards and Craven, 2017). Regionally, sea levels in Ireland had stabilised by the 20th century after which

sea levels began to increase again due to anthropogenic-induced warming (Masson-Delmotte et al., 2021). The importance of climate warming and SLR in Ireland has been emphasized by a number of authors: Devoy (2008) discussed the risks of extreme climatic events and the ways in which Ireland should be prepared for them, Camaro Garcia et al. (2021) states that



satellite observations show sea levels rising around Ireland at a rate of 2–3 mm/yr, in line with global averages for the early
21st century. However, the raw tide gauge record at Dublin Port shows a rate of 0.3 mm/yr in sea level from 1938 to 2000

(DCC, 2005), much lower than the global average.

A number of authors have investigated trends in the Dublin Port tide gauge prior to the year 2000, finding similarly low rates
of change. Carter (1982) investigated the Dublin Port record using tide gauge measurements and reported a rising trend of 0.6
mm/yr before 1961 and a falling trend of -0.3 mm/yr from then until 1980. Woodworth et al. (1991, 1999) estimated trends
of 0.17 mm/yr ($\pm$ 0.35) from 1938–1988, and 0.23 mm/yr from 1938–1996. In stark contrast to these low rates of SLR, the

recently published climate change action plan for 2019–2024 by Dublin City Council (DCC, 2017), reports 6-7 mm/yr SLR
between the years 2000 and 2016. This rate is approximately double that of global mean sea level rise (Nerem et al., 2018) and
particularly surprising given the earlier rates of rise in Dublin were much lower than global mean sea level rise over similar
periods (Dangendorf et al., 2017).

The goal of this paper is to further investigate the sea level trend in Dublin Port through careful assembly and quality control

of the available data and by comparing sea level records collected from nearby tide gauges. We find that Dublin's available
sea level measurements are not aligned with those of nearby tide gauges and thus need further processing. In particular, we
find problems with the mean high water measurements which indicate a drift over time. After adjustment for the drift and for
other atmospheric factors, we find that the sea level record, at least for the 21st century, matches other local tide gauges to a far
higher degree. This allows us to estimate more reliable measurements of sea level rise for the urban area of Dublin Port.

The rest of this paper is organised as follows. Section 2 explains how the sea level dataset for Dublin Port is reconciled from
various sources. Section 3 discusses the quality check and calibration procedures done on the reconciled dataset. Section 4
discusses SLR rates at Dublin Port. Some issues and suggestions pertaining to SLR analysis at Dublin Port are discussed in
section 5. Finally, the important findings of this study are summarised in section 6.

## 2   Data collation for Dublin Port

We compiled mean high and low water, mean tide level and, where available, mean sea level for Dublin Port from 1938 to 2018
from the following sources:

1. Annual high and low water from Woodworth et al. (1991) for the period 1938 to 1988 from annual tabulations made by
   the Dublin Port Authority (hereafter the Port Authority Annual Dataset). Data from 1938-77 are relative to Port Datum,
   which is 0.436 m above the Ordnance Survey Datum Dublin (Poolbeg Datum) and data from 1978-1988 are relative to

the Lowest Astronomical Tide (LAT).

2. Monthly values of mean high water (MHW) and mean low water (MLW) for the period of 1987–2001 (hereafter the
   Port Authority Monthly Dataset), which were digitized as part of this study. Quality control measures for the digitisation
   included automatic checking of the calculated and recorded mean levels. Overlapping years with the Port Authority An-





nual Dataset were used to evaluate agreement with the older dataset. There was approximately 1 cm difference between

high water values. Low water values agreed to within millimetric accuracy. Data are reported relative to LAT.

3. High frequency (10 minute) data supplied by the Permanent Service for Mean Sea Level (PSMSL, 2020; Holgate et al., 2013) for the period 2001-2009. These data were provided to PSMSL by the Harbour Master in Dublin Port following a change in responsible authority in 2001 (hereafter the Harbourmaster Dataset). Data have a low vertical resolution of 0.1 m. Data are reported relative to LAT.

4. High frequency (5 minute) data for the period 2006–2017 from the Irish National Tide Gauge Network (NTGN), which is maintained by the Irish Marine Institute (hereafter the NTGN dataset)(IMI, 2019). Data are available relative to Ordnance Datum Malin (ODM) and LAT. All NTGN data are taken relative to ODM in this study.

5. High water levels for the period 1968–2013. These were digitised as part of an unpublished MSc thesis of A. Greene (hereafter the Greene dataset) and are published here for the first time. The Greene dataset for the period 1968-1982 was

transcribed from photographs of tidal charts from which the high water values can be read. During 1983-2003 the data were in the format of hard copy tidal charts. The hard copies consisted of 3 large A3 books. The remaining data from 2003–2013 existed in digital format from which high waters could be derived.

With particular respect to the Greene dataset, prior to the availability of digital data in 2003, the high water values for each day were extracted from the tidal charts. This was completed by the generation of tables for each year, with two available

cells for each day, these values were read off and inputted into the designated cell. The data from the period 1968-1976 was converted from feet and inches to meters. Data post 1998 were already digitised at 15 minute time intervals, post 2004 this data's frequency increased to 1 minute intervals. To locate the two high tides, each month was split into days, sorted with the highest value being extracted for high tide 1. The second-high tide occurred between 12 and 13 hours after the first high tide, therefore by using the time component within the dataset, this value was extracted. A summary of the datasets is shown in

Table 1.

**Table 1. Details of the datasets collated to form a complete sea level record for Dublin Port**

| Dataset name | Duration | Sampling frequency | Provider |
|---|---|---|---|
| Port Authority | 1938-1977 | Annual | Woodworth et al. (1991) |
| Port Authority | 1978-1988 | Monthly | Woodworth et al. (1991) |
| Harbourmaster | 2002-2009 | 10-minute | PSMSL |
| NTGN | 2006-2019 | 5-minute | Marine Institute |
| Greene | 1968-2013 | Twice daily | This study |

Difficulties in merging the Dublin Port datasets arose from differing datum definitions. For both Port Authority datasets the tabulated annual and monthly data are relative to the same datum. These data have the same source and therefore agreement is expected. Four years of overlap exist between the Harbourmaster dataset and the NTGN dataset from 2006 to 2009. The



Harbourmaster dataset is relative to LAT datum; NTGN data are relative to ODM, with a value of 2.599 m between these
datums. There is a systematic underestimation of 0.044 m in MHW and MLW values in the Harbourmaster dataset due to its
lower vertical resolution of 0.1 m, determined from the overlap with the NTGN data.

While no overlaps exist between the Harbourmaster dataset and the Port Authority dataset, the Greene dataset overlaps the
Port Authority, Harbourmaster, and NTGN datasets. Figure 1 shows MHW from the monthly Port Authority, Harbourmaster,
NTGN, and Greene datasets. There is good agreement between the data indicative of consistent datums. We find a residual
0.008m difference between the Greene dataset and the Port Authority Monthly dataset and the NTGN Dataset. We thus add 8
mm to the Greene dataset as the final datum adjustment.

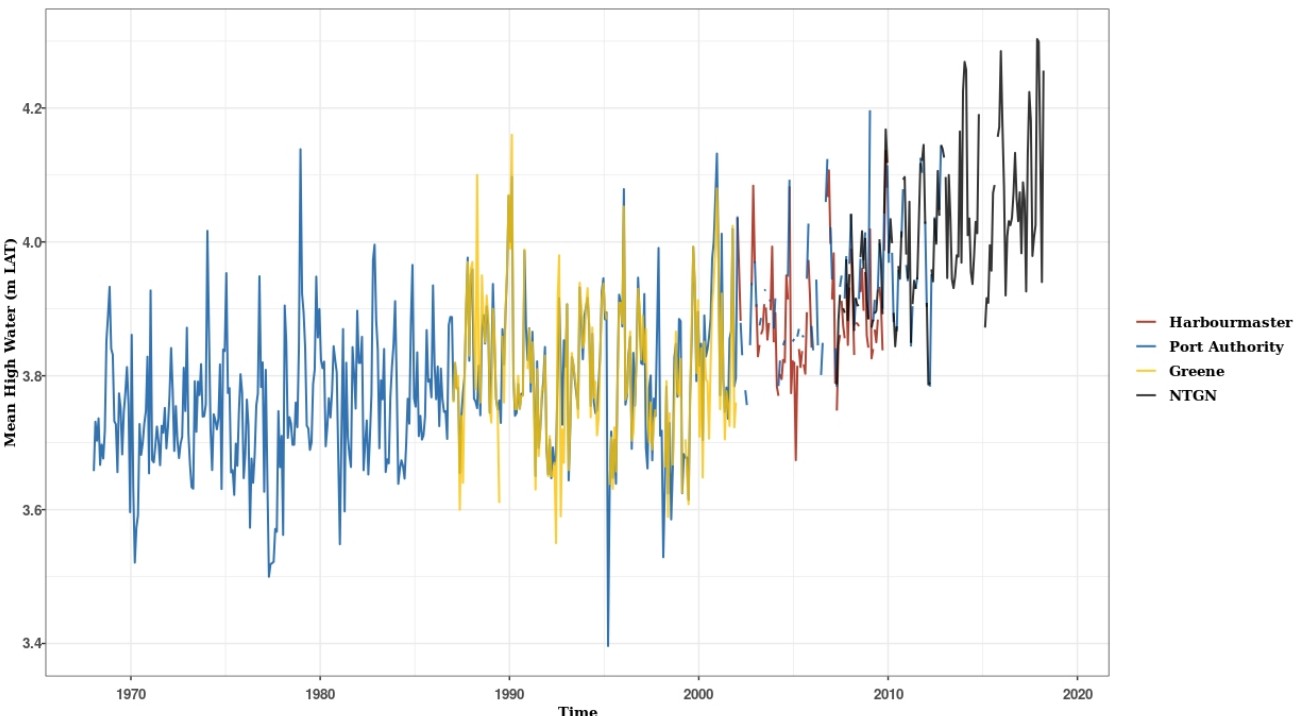

**Figure 1.** Monthly MHW in Dublin from the Port Authority, Greene, Harbourmaster, and NTGN datasets. Good agreement is found between
the records indicating consistent datum definition, with only a small adjustment of 0.008 m to the Greene dataset for complete reconciliation.

## 3 Reconciliation of Dublin Port against nearby tide gauges

We now use our newly merged dataset (hereafter the merged Dublin Port dataset) including monthly MLW, MTL, MSL and
MHW measurements between 1938 and 2019. Due to the systematic underestimation of MHW in the Harbourmaster dataset,
we supplement the monthly averages of the high waters from the Greene dataset for the period 2002 to 2006 and calculate
mean low waters by combining this mean high water with the mean tide level. Full tidal data are not available prior to 2003 and





only tabulated MTL is available. Based on calculations from the NTGN dataset, an adjustment of 0.049 m is added to MTL for conversion to MSL, in line with the results of Woodworth (2017) regarding the difference between MTL and MSL.

In order to check the reliability of the merged Dublin Port dataset, we compare with two different nearby tide gauges (maximum distance 60 km) at Howth Harbour and Arklow, and two other tide gauges at Newlyn in the UK and Brest in France. Figure 2 shows the locations of the tide gauges. The recordings, when taken together, have different time spans and different sampling frequencies. Table 2 provides the details of the datasets.

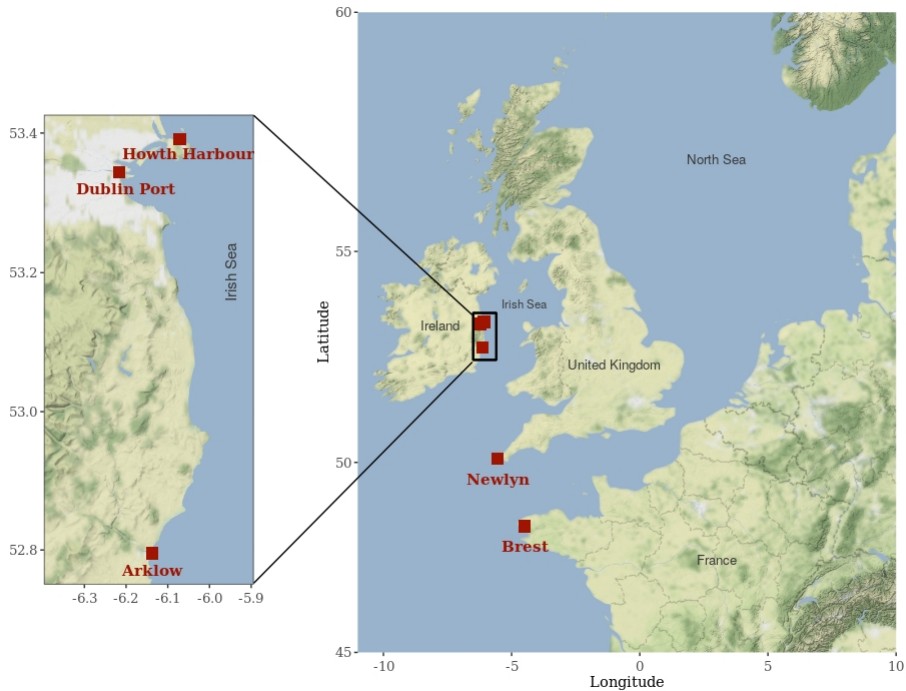

**Figure 2.** Locations referred to in this study: Dublin Port, Howth Harbour, Arklow, Brest and Newlyn.

**Table 2. Details of the four datasets used to compare with Dublin Port**

| Dataset | Duration | Sampling frequency | Provider |
|---------|----------|--------------------|----------|
| Arklow | 2003-2019 | 15-minute | Office of Public Works |
| Howth harbour | 2007-2019 | 6-minute | Marine Institute |
| Brest | 1938-2016 | Annual | PSMSL |
| Newlyn | 1938-2016 | Annual | PSMSL |

PSMSL provides both Monthly and Annual data. We use only annual data that matches our data resolution.

For the Arklow and Howth Harbour datasets, we first aggregate the values up to daily and monthly level for MLW, MSL and MHW.





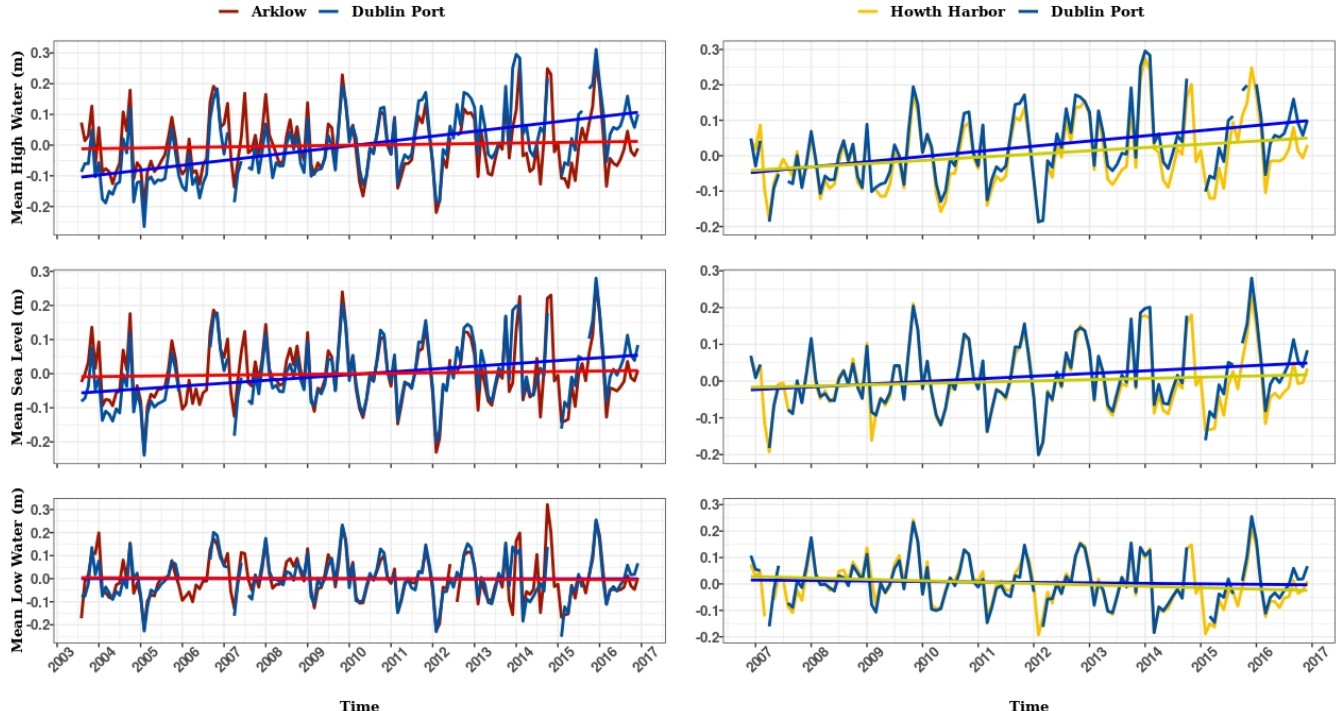

**Figure 3.** MHW, MSL and MLW monthly values of Dublin Port against Arklow (left panels) and Howth Harbour (right panels) with linear trends for each. The MLW linear trends between Dublin Port and Arklow and between Dublin Port and Howth Harbour are in good agreement. However this is not the case with the linear trends for MSL and MHW when comparing Dublin Port to the two other locations.

**Table 3. Differences between rates in MLW, MSL and MHW for Dublin Port compared to Arklow and Howth Harbour. Large values with small standard errors indicate a significantly higher rate at Dublin Port.**

| Locations | MLW (mm/yr) | MSL (mm/yr) | MHW (mm/yr) |
|---|---|---|---|
| Dublin Port - Arklow | 0.6 (±2.76) | 6.94 (±2.63) | 13.99 (±2.74) |
| Dublin Port - Howth Harbour | 1.51 (±4.03) | 3.85 (±3.91) | 5.8 (±4.25) |

Figure 3 and Table 3 demonstrate that rates of SLR in MSL and MHW are significantly higher in Dublin Port than in Arklow or Howth Harbour. A possible cause of the issue is the malfunction of the tide gauge in measuring the high water levels due to drift. A Druck pressure transducer was utilised at Dublin Port (Murphy et al., 2003) and has the potential to exhibit drift proportional to the height of the water column. Starting with the MLW values at Dublin Port, which compare better with nearby gauges, we construct a baseline to correct the MSL values. To do this, we create a regression model that estimates MSL given
MLW from older Dublin Port measurements. We then use the predictions from this model to estimate MSL at Dublin Port for the more recent period.



To find the period of time over which to train the regression model, we use a change point model (Carlin et al., 1992) that takes the absolute difference between MSL values of Dublin Port and Newlyn in the UK as the inputs. Details of the change point model are discussed in the Appendix. According to the model years 1939-1976 are chosen as training data.

Figure 4 shows the comparison of Dublin Port MSL data with that of Newlyn and Brest in France. These two are selected due to the relative completeness and integrity of their record and their proximity to Dublin. According to the figure, there is strong agreement between stations for MSL in the period 1939 to 1976. After 1976 the level of agreement deteriorates which is consistent with the change point model result.

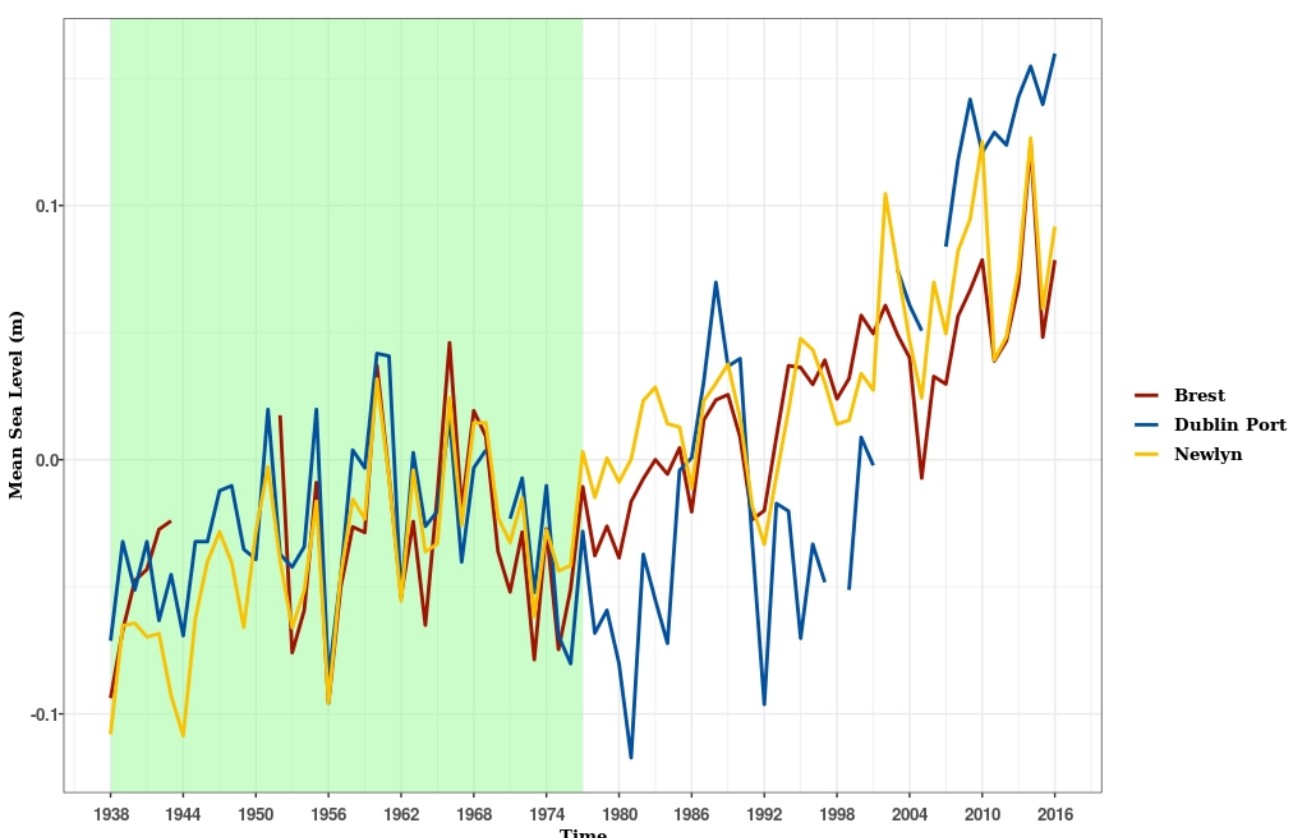

**Figure 4.** Yearly MSL values for Dublin Port, Newlyn and Brest. The green area shows our chosen time period during which there is good agreement between Dublin Port and the other sites.

We estimate the yearly MSL using Bayesian multivariate linear regression. Our model consists of an intercept, a fixed effect

on annual MLWs, and a harmonic function with a period of 18.6 years to model the lunar nodal cycle, and a period of 4.4 years to account for the 8.85 year cycle of lunar perigee (Haigh et al., 2011). The model formulation is as follows:





$$(MSL)_t \sim N(\mu_t, \sigma^2)$$

$$\mu_t = \beta_0 + \beta_1 (MLW)_t + \beta_2 \cos(\omega_1 t) + \beta_3 \sin(\omega_1 t) + \beta_4 \cos(\omega_2 t) + \beta_5 \sin(\omega_2 t)$$

$$\text{with } \omega_1 = \frac{2\pi}{18.61}, \ \omega_2 = \frac{2\pi}{4.4}$$

where $(MSL)_t$ is MSL in year $t$, $\mu_t$ is the mean process, $\sigma^2$ is the residual variance, $\beta_0$ is the intercept, $\beta_1$ is the MLW coefficient, $\beta_2$ and $\beta_3$ are the amplitudes of the cosine and sine functions of the 18.6-year lunar nodal modulation respectively, and $\beta_4$ and $\beta_5$ are the amplitudes of the cosine and sine functions of the 4.4-year modulation respectively.

We fitted the model using the JAGS software (Denwood, 2016) and R (R Core Team, 2020) and used 3 Markov Chain Monte Carlo chains, 2000 iterations per chain with 1000 as burn-in and a thinning value of 1. Convergence was assessed using the R-hat diagnostic (Brooks and Gelman, 1998; Gelman and Rubin, 1992). All R-hat values associated with $\beta$ and $\sigma$ were close to 1 so the model was assumed to be sampling from the posterior distribution. The new predicted MSL and the old raw MSL values, together with the yearly MSL values of Arklow and Howth Harbour, are shown in Figure 5. This figure demonstrates
that the newly modelled Dublin Port yearly MSL data are changed only slightly between 1939 and 2001. After 2001 we can see a clear gap between the old (red) and new (blue) versions with the new corrected data exhibiting superior agreement with the Arklow and Howth Harbour records. We note again here that these adjacent records were not used in the creation of the new Dublin Port data so are independent validation on our approach. A further justification of our adjustment approach can be seen in that the period 1978 to 2001 was not included in our calibration dataset, but provides a relatively small adjustment to
MSL

## 4   Sea level rise at Dublin Port and nearby gauges

We now use the Dublin Port corrected data to calculate rates of sea level rise. We use the yearly MSL data from Brest and Newlyn for comparison. We first removed the atmospheric effects following Diabaté et al. (2021) and Frederikse et al. (2017). Atmospheric data are accessed via the RNCEP package (Kemp et al., 2012) in the R programming language which accesses the
National Centers for Environmental Prediction (NCEP)/National Center for Atmospheric Research (NCAR) and Department of Energy Reanalysis I & II datasets (Kalnay et al., 1996; Kanamitsu et al., 2002). Figure 6 shows the atmospherically corrected MSL data of Dublin Port, Brest and Newlyn superimposed for comparison.

To calculate the SLR rates, as before, we use a Bayesian multivariate linear regression including an intercept, a linear trend, and a harmonic function with periods of 18.6 years and 4.4 years. The model is fitted in JAGS with the same settings and convergence requirements as previously described. We write the model as:

$$(MSL)_t \sim N(\mu_t, {\sigma'}_t^2 + \sigma^2)$$

$$\mu_t = \beta_0 + \beta_1 t + \beta_2 \cos(\omega_1 t) + \beta_3 \sin(\omega_1 t) + \beta_4 \cos(\omega_2 t) + \beta_5 \sin(\omega_2 t)$$

$$\text{with } \omega_1 = \frac{2\pi}{18.61}, \ \omega_2 = \frac{2\pi}{4.4}$$





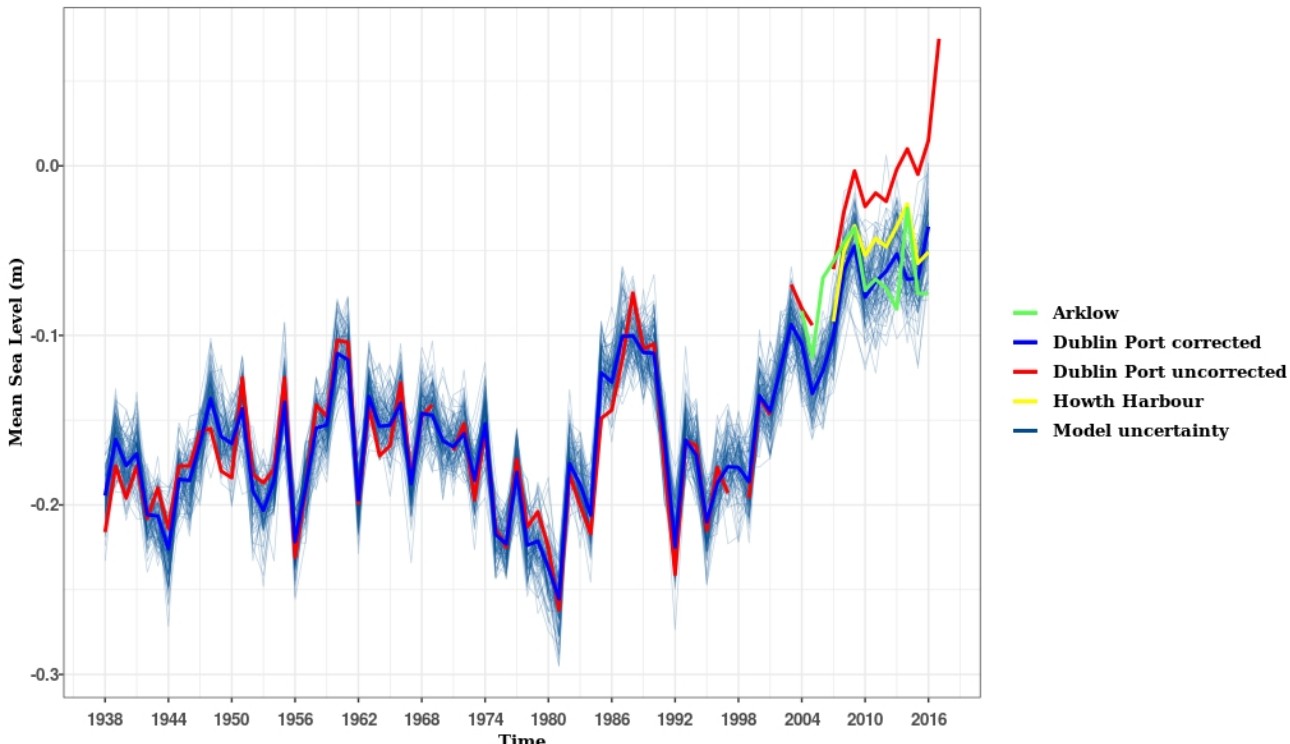

**Figure 5.** The uncorrected and corrected yearly MSL values of Dublin Port, with yearly MSL values of Arklow and Howth Harbour for comparison. The newly corrected Dublin Port MSL values lie much closer to the neighbouring tide gauges. The faded lines in the background show posterior samples from the model and an indication of model uncertainty.

where $(MSL)_t$ is MSL at time $t$, $\mu_t$ is the mean process, $\sigma'^2_t$ is the fixed variance at time $t$ extracted from the posterior distribution of the calibration model to account for the uncertainty in modelling the MSL introduced in the previous section, $\sigma^2$

is the residual variance, $\beta_0$ is the intercept, $\beta_1$ is the rate in mm/yr, $\beta_2$ and $\beta_3$ are the amplitudes of the cosine and sine functions of the 18.6-year lunar nodal modulation respectively, and $\beta_4$ and $\beta_5$ are the amplitudes of the cosine and sine functions of the 4.4-year modulation respectively. We use the same approach (but without the fixed measurement error) for estimating the rates of rise at Brest and Newlyn.

The estimated rates with their associated 95% posterior credible intervals are given in Table 4 which shows that, between

1953 and 2016, the rate of SLR at Dublin Port has mean estimate of 1.08 mm/yr, consistent with the estimated rate of 1.06 mm/yr at Brest and that of 1.4 mm/yr at Newlyn. However in more recent years, specifically between 1997 and 2016, Dublin has experienced a greater SLR of 6.48 mm/yr, larger than that of 2.59 mm/yr at Brest, and 3.69 mm/yr at Newlyn. Figure 6 also suggests that sea level in Dublin Port has experienced larger decadal fluctuations and is not as secular as the sea level at the two other locations.





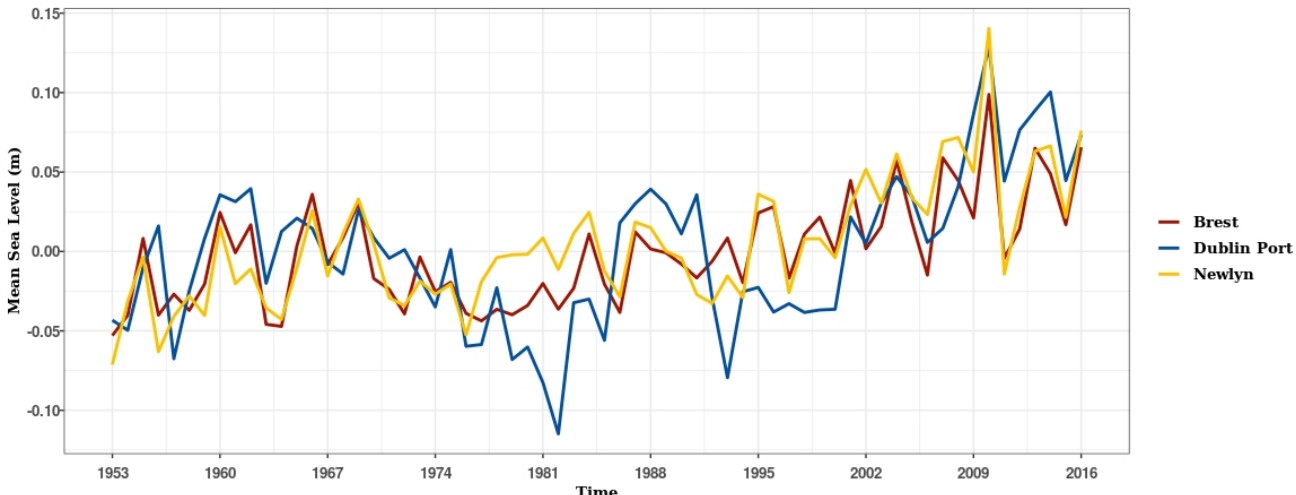

**Figure 6.** New yearly MSL values of Dublin Port and yearly MSL values of Brest and Newlyn, with atmospheric effects removed, between 1953 and 2016.

**Table 4. Estimated rates of SLR at Dublin Port, Newlyn and Brest, with 95% credible intervals.**

| Location | 1953-2016 (mm/yr) | 1975-1985 (mm/yr) | 1986-1996 (mm/yr) | 1997-2016 (mm/yr) |
|---|---|---|---|---|
| Dublin Port | 1.08 (0.62, 1.55) | 3.41 (-2.32, 9.16) | -9.8 (-15.7, -4) | 6.48 (4.22, 8.8) |
| Newlyn | 1.40 (1.09, 1.71) | 4.06 (0.91, 7.24) | 1.57 (-4.23, 7.24) | 3.69 (0.28, 7.07) |
| Brest | 1.06 (0.76, 1.36) | 2.55 (0.14, 5) | 2.94 (-0.27, 6.33) | 2.59 (-0.31, 5.56) |

## 5 Discussion

Taken over the full time period of observations, 1953 to 2016, the estimated sea level rise of 1.08 mm/yr in Dublin is consistent with that of Brest and Newlyn, both located at the western European coastline. The rates of rise for earlier periods are less than 1.08 mm/yr (Carter, 1982; Woodworth et al., 1991) are consistent with the findings here and were lower due to the decades of larger sea level rise and variability (1980s, 2000s) not being included in the trend estimation. Elsewhere in Ireland Orford et al. (2006) investigated tide gauge records in Malin Head (1958 - 1998) and Belfast harbour (1918 - 2002) where they reported substantial annual variation for both sites with overall negative trends of -0.2 mm/yr for Belfast and -0.16 mm/yr for Malin Head. Both Belfast and Malin Head, being in the north of the country, are in regions of Glacial Isostatic Uplift, which will reduce relative sea level rise there (Bradley et al., 2011). However, Dublin is in a region of neutral Glacial Isostatic Uplift so these long term effects of post-glacial land motion should be negligible and hence greater consistency with the global figure is expected and ultimately found.

More surprising is the large decadal variability revealed. This study has found a rate of sea level rise for Dublin of 3.41 mm/yr for the period 1975–1985 followed by a negative sea level trend in the next decade (-9.8 mm/yr during 1986-1996) and





again a rise of 6.48 mm/yr for the period 1997-2016. Ireland sits on the edge of the Atlantic, greatly influenced by decadal patterns of climate such as Atlantic Multidecadal Oscillation (AMO) (McCarthy et al., 2015) and decadal variations in the

strength of the North Atlantic Oscillation (McCarthy et al., 2018). Similar patterns of decadal variability in sea level to Dublin were noted in Belfast by Orford et al. (2015) and linked to decadal variation of the North Atlantic Oscillation. This would seem a likely explanation for similar patterns in Dublin. However, a full investigation of the causes of decadal variability in Dublin sea level remains for future investigation.

Comparisons of MLW, MSL and MHW recordings at Dublin Port suggest a possible issue with the observation of high water

levels. Our model recreated the MSL and showed that there is good agreement between the observed MSL and the modelled MSL for the period 1939-2001. However after 2001 there is considerable divergence. Our analysis shows that the modelled MSL is more consistent with the data collected by the nearby tide gauges and also at the farther sites in Newlyn and Brest. This suggests that the malfunction probably started during or after the year 2002. We would consider the Howth Harbour sea level record, alongside the modelled MSL data created in our study, as more reliable dataset for future analysis of sea level in

Dublin Bay compared to Dublin Port's dataset.

Dublin City Council have recently increased the coastal defences in Dublin, allowing for between 40 and 65 cm of mean sea level rise (O'Connell, 2019). Projections of sea level rise for Dublin, based on UKCP18 (Fung et al., 2018), depend heavily on greenhouse gas emissions trajectories. By 2100, Dublin mean sea level is projected to rise by 0.6 m at the 50th percentile (1.0 m at the 95th percentile) under an RCP8.5 scenario and 0.3 m at the 50th percentile level (0.6 m at the 95th percentile level)

under an RCP2.6 scenario. These projections do not simulate the decadal scale variability reported here, similar to many other decadal climate phenomena. Understanding the origin and duration of the decadal fluctuations of mean sea level in Dublin is crucial for preparation and defence of Ireland's capital city in the coming decades.

## 6  Conclusions

We have collated multiple sources of tide gauge data for Dublin Port, and subsequently corrected them for bias in the MHW

level. We have then shown that these corrected MSL measurements agree with both Howth Harbour and Arklow to a far higher degree than the raw data. A longer term comparison with Brest and Newlyn also indicates overall agreement. There remains a difference between the data during the 1970s and 1980s where a large cyclic disparity in Dublin contrasts with the other two records. Our final adjusted dataset estimated the rate of SLR to be 1.08 mm/yr between 1953 and 2016, and 6.48 mm/yr between 1997 and 2016 at Dublin Port.

The work we present here is part of a broader aim to improve sea level records in Ireland through the multi-centre Aigéin, Aeráid, agus athrú Atlantaigh (A4) project. A recent example is that of the now corrected tide gauge record in Cork City (Pugh et al., 2021). We hope to report elsewhere on further records which may provide a fuller picture of SLR in Ireland.




## Appendix A

To identify the period of data to use in calibrating the Dublin Port MSL and MLW values (see Figure 4), we use a linear
regression change point model (Carlin et al., 1992). The model we use can be formulated as follows

$$y_t \sim N(\mu_t, \sigma^2)$$

$$\mu_t = \alpha_1 + \alpha_2 \times u(t - t_c)$$

where $y_t$ is the absolute difference between the measured mean sea level at Dublin Port and Newlyn at time $t$ ($t = 1, 2, \ldots, T$).
We assume $y_t$ to be normally distributed with mean $\mu_t$ and variance $\sigma^2$. The mean is set to $\alpha_1$ if $t < t_c$ and $\alpha_1 + \alpha_2$ otherwise,
and $t_c$ is the time of the change point. The function $u(t)$ is the unit step function. We used vague prior distributions for all
parameters:

$$\alpha_1 \sim N(0, 100)$$

$$\alpha_2 \sim N(0, 100)$$

$$t_c \sim Unif(1938, 2016)$$

$$\sigma^2 \sim Unif(0, 100)$$

The model output is shown in figure A1. The vertical red line indicates the year of the change point and so marks the end of
the calibration period as described in Section 3.





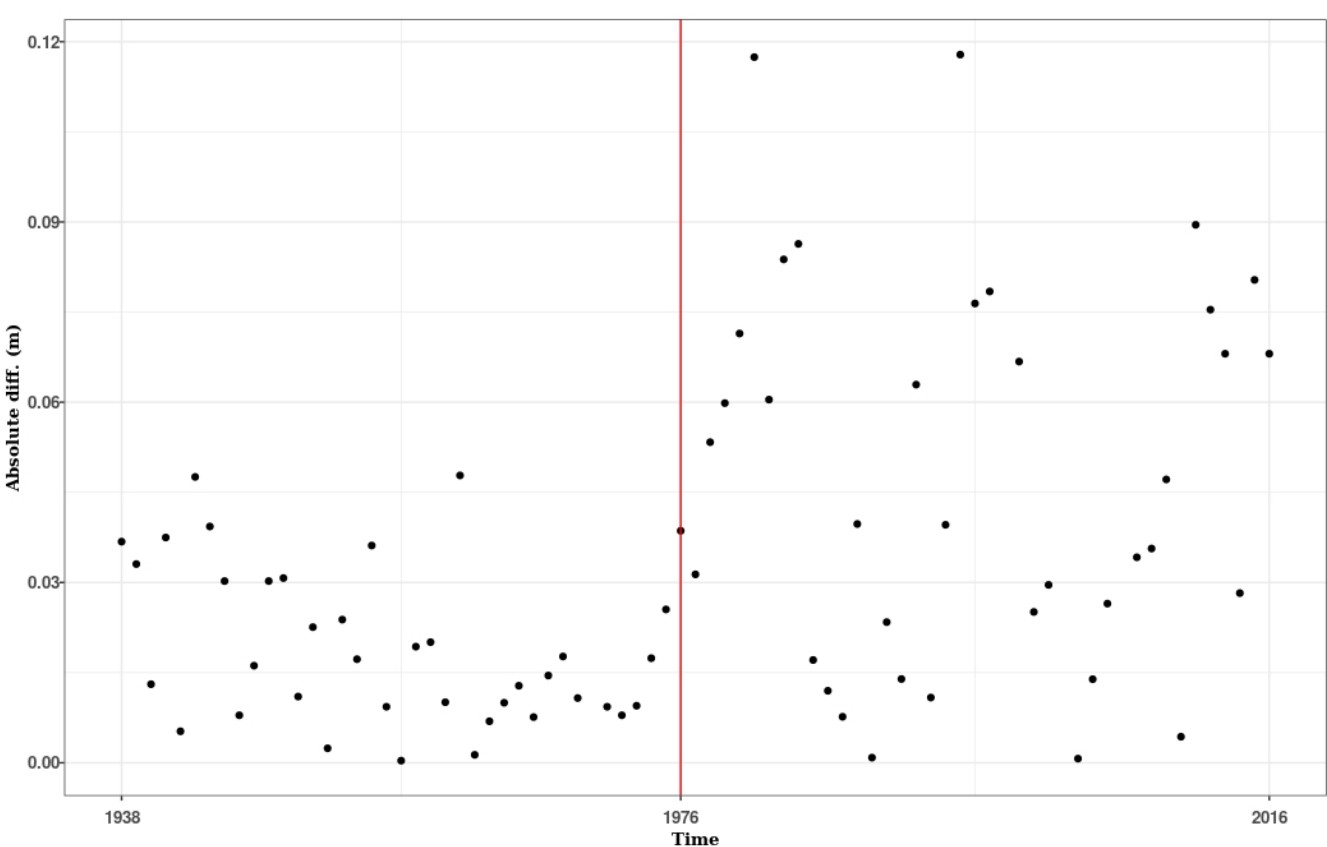

**Figure A1.** The absolute difference between Dublin Port and Newlyn from 1938 to 2016. The mean posterior estimated change point time given by the model is indicated by the vertical red line at year 1976.



*Code availability.* The codes used in this study are available on Github (https://github.com/Aminsn/Dublin-Bay-SLR)

*Data availability.* Dublin Port annual mean sea level data (corrected as a result of this study for measurement bias) from 1938 to 2016 are available in CSV format (https://doi.org/10.15129/5d28213e-8f9f-402a-b550-fc588518cb8b.)

*Competing interests.* The authors declare that they have no conflict of interest.

*Acknowledgements.* Amin Shoari Nejad's work was supported by a Science Foundation Ireland Investigator Award grant 16/IA/4520. Andrew Parnell's work was supported by a Science Foundation Ireland Career Development Award (17/CDA/4695), a research centre award (12/RC/2289-P2), an investigator award (16/IA/4520), and a Marine Research Programme funded by the Irish Government, co-financed by the European Regional Development Fund (Grant-Aid Agreement No. PBA/CC/18/01). Gerard McCarthy is supported through the A4
project (Grant-Aid Agreement No. PBA/18/CC/01) is carried out with the support of the Marine Institute and the Marine Research Programme 2014-2020, co-financed under the European Regional Development Fund. For the purpose of Open Access, the author has applied a CC BY public copyright licence to any Author Accepted Manuscript version arising from this submission.





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
