# Peer review of "A newly reconciled data set for identifying sea level rise and variability in Dublin Bay"

_Ocean Science, 2021_

## Referee Comment (RC2)

**Review of manuscript:** os-2021-92
*A newly reconciled data set for identifying sea level rise and variability in Dublin Bay*

**General comments:**

The authors have produced a new record for sea level (MWH, MLW and MSL) for the Dublin Port region. They have looked into the outstanding issue of why the Dublin tide gauge (or longer term) rate of sea level rise is different to other records. This was an important issue to investigate, given the wide range of possible rates that have been published.

They combined multiple published and unpublished datasets for the region and identified an issue the MHW and MSL. They corrected for this, using the MLW records to produce an updated MSL record applying Bayesian multivariate linear regression.
They compared the corrected MSL records for Dublin Port to two regional tide-gauges from Brest and Newlyn, to put the updated rates from Dublin into a more regional context.

I have a few questions regarding the figures, mostly are minor to clarify a few things. I have few more detailed questions, to clarify parts of the manuscript, but only minor points.

**1: Section 4**: Prior to calculating the MSL rates line 140 states " *yearly MSL from Brest and Newlyn .. removed atmospheric effects following"*. If to use the MSL from Brest +Newlyn the atmospheric effects need to be removed; why was this not removed in Section 3 when estimating the Mean sea level for Dublin Port.

**2: Section 5 and Section 6:** Unless the journal requires both discussion and conclusion section: these could be combined as "Discussion +Conclusions"
In the introduction the authors refer to the difference between satellite observations (*line, 23 " sea level rising at a rate of 2-3 mm/yr*), Dublin City council rate (*line 30: reports a 6-7 mm/yr SLR between years 2000-2016)* and the rate from previous published studies. Could the authors refer to these differences in the discussion or conclusions. Why are the satellite records different, for example?

**Minor questions:**
**1:Section 1:** *Line 34-40 "find problems with the MHW measurements which indicate a drift over time"* From reading section 3, I assume this "drift correction" was calculated using the Bayesian multivariate linear regression. I think it would to mention this.

*2: Data collection for Dublin Port:*
*Line 45 - 66. We complied MHW... and, where available, mean sea level for Dublin Port.* Do all five datasets have data for MHW, MLW, MTL and MSL? If not, could you add to table 1 which do/do not. From Figure 1, I assume they all provide MHW, but which ones do not record MSL?

**3: Section 4:** Perhaps rename "*rates of sea level rise at Dublin Port and nearby tide-gauges*". Just to make it clearer the differences of this section to Section 3.

**Specific comments about figures**

**1: Table 1**: Can you add a column for the different datums used for each of the datasets, prior to processing?  On line 76 "*Difficulties in merging the Dublin Port datasets arose from the differing datum definitions*"
It would be helpful to have Table 1 and Table 2 combined - This is useful to compare the different durations of each record.  Also can you check the range of each data record through the paper as the text and table often do not match or are consistent.
     -Port Authority Annual: duration: 1938-1977. *Line 46: " refer to as 1938 to 1988*"
     -Port Authority Monthly: duration 1978-1988: *Lines 45- 50.* "*The Monthly Port Authority data for the period 1987-2001*". Is this combined with the first dataset refer to in as Port Authority Annual
     -Harbourmaster: duration: 2002-2009. Line 56: *"data supplied by the PMSL for the period 2001-2009".*
     -NTGN: duration: 2006-2019. Line 60: *"data for the period 2006-2017"*

**2: Figure1:** The length of the records shown do not correspond to the length of the records in Table.1. For example: Port Authority (blue) ~ 1968 to 2011? This may be a plotting problem or the resolution of the figure.

***3: Line 89****: "we now use our newly merged dataset (..) MSL and MHW from 1938 and 2019.* As you refer to the period 1938 - 2019, why is Figure1 showing the MHW from 1968 to 2019?

**4: Table 2 and Figure 3:**  Can you clarify why the time axis is only to 2017 (but the records extend to 2019?)

---

## Author Comment (AC1)

Table 1- Details of the Tide gauges installed at Dublin port.

| Gauge type | Start | End | Reference |
|---|---|---|---|
| Float Gauge | 1938 | 2001 | Original charts supplied by Dublin City Council indicate float gauge in operation from records beginning to 2003. |
| Druck Pressure Transducer | 2001 | 2016 | Stated sensor type in Murphy et al., 2003. This sensor began to malfunction in 2017 and was replaced in 2018. |
| OTT Bubbler | 2018 | - | New sensor installed in 2018. Data not used in this study. |

Table 2- Details of the datasets collated to form a complete sea level record for Dublin Port.

| Dataset name | Duration | Sampling frequency | Provider |
|---|---|---|---|
| Port Authority | 1938-2001 | Annual | Woodworth et al. (1991), PSMSL |
| Port Authority | 1978-2001 | Monthly | Woodworth et al. (1991), PSMSL |
| Harbourmaster | 2002-2009 | 10-minute | PSMSL |
| NTGN | 2006-2019 | 5-minute | Marine Institute |
| Greene | 1968-2013 | Twice daily | This study |

Table 3- Dublin's SLR rate during different periods with and without Perigean.

| 1953-2016 (with perigean) | 1953-2016 (without perigean) | 1997-2016 (with perigean) | 1997-2016 (without perigean) |
|---|---|---|---|
| 1.08 (95% CI: 0.62 – 1.55) | 1.1 (95% CI: 0.57 –1.57) | 6.48 (95% CI: 4.22 – 8.8) | 7.07 (95% CI: 5.15 –8.93) |

[Figure]

Figure 1. MHW, MSL and MLW monthly values of Dublin Port against Arklow (left panels) and Howth Harbour (right panels)

---

## Author Response (AR1)

Dear Professor Huthnance,

Many thanks for considering our paper titled 'A newly reconciled data set for identifying sea level rise and variability in Dublin Bay'. We have revised the manuscript following the referee's comments (listed on the next page) and tried addressing all of the queries during the discussion period.

We hope that you now find the paper suitable for publication. If you have any further queries please do not hesitate to get in contact with me.

Yours sincerely,

Amin Shoari Nejad

**Comments from Prof. Woodworth:**

> One major comment is that there is no technical information or history provided on each gauge.

We added a new table (table-4 in the revised manuscript) including the history of the gauges at Dublin Port.

> A second comment is that Table 1 is missing a line?

We fixed the table 1 to incldue all the necessary information.

> A third commment is that on line 1 of the abstract it says the new composite record for Dublin will be for 1938-2016, but from Table 1 we see there is Dublin data to 2019. It looks like the 2016 constraint comes from Newlyn and Brest data being only to 2016 (Table 3). That may have been the case when the first draft of this paper was written but there are data to 2019 for both now so the new Dublin record could go to 2019.

Lines 94-97 in the revised manuscript explain why we removed data post 2016.

> A fourth comment relates to the regressions at lines 116 and 140. I think the former is fair enough (although see the additional reference mentioned below). The point is that the MLW will contain a nodal component mostly due to changes in the tide (the roughly 3.7 percent of the M2 tide) and also a small nodal component in MSL. It will also have a perigean component due to the tide. So ok. But the latter is not so reasonable. MSL will have a nodal component (seeWoodworth, 2012) but it will be very small. So, instead of determining the true nodal amount, your fit over 38 years will simply pick out noise from the much larger interannual variability in the ocean circulation. In addition, there is no tidal basis for including a perigean component in a parameterisation of MSL. So I think this equation is not

> reasonable. Anyway, you don't discuss the determined beta2-5. So I would also make a simple regression with beta0 and beta1 and check if those parameters are similar to what you have here then use them instead. (If they are different then you have a problem).

We fitted the model again, both with and without the perigean, and the results changed only slightly. We removed the perigean in the revised manuscript and adjusted all the assoicated figures as a result of this change.

We tried addressing all of the minor comments mentioned by Prof. Woodworth and all of the associated changes are highlighted in the submitted "track-changes" file.

**Comments from Dr. Bradley:**

> Prior to calculating the MSL rates line 140 states " yearly MSL from Brest and Newlyn .. removed atmospheric effects following". If to use the MSL from Brest +Newlyn the atmospheric effects need to be removed; why was this not removed in Section 3 when estimating the Mean sea level for Dublin Port.

We explained why we chose this approach in lines 137-138 of the revised manuscript.

> Section 5 and Section 6: Unless the journal requires both discussion and conclusion section: these could be combined as "Discussion +Conclusions" In the introduction the authors refer to the difference between satellite observations (line, 23 " sea level rising at a rate of 2-3 mm/yr), Dublin City council rate (line 30: reports a 6-7 mm/yr SLR between years 2000-2016) and the rate from previous published studies. Could the authors refer to these differences in the discussion or conclusions. Why are the satellite records different, for example?

If the editor believes that there is no problem with merging the Discussion and Conclusion sections, we have no objection to follow the reviewer's suggestion.

Regarding the rates, in line 23 of the revised manuscript we emphaised that satellite observations are associated with the open ocean rate.

> 1:Section 1: Line 34-40 "find problems with the MHW measurements which indicate a drift over time" From reading section 3, I assume this "drift correction" was calculated using the Bayesian multivariate linear regression. I think I would to mention this.

We added a sentence (lines 37-38 in the revised manuscript) clarifying this issue.

> Data collection for Dublin Port: Line 45 - 66. We complied MHW... and, where available, mean sea level for Dublin Port. Do all five datasets have data for MHW, MLW, MTL and MSL? If not, could you add to table 1 which do/do not. From Figure 1, I assume they all provide MHW, but which ones do not record MSL?

In the revised manuscript, we added a new colomn to Table -1 called "Variables" including the requested information.

> Section 4: Perhaps rename "rates of sea level rise at Dublin Port and nearby tide-gauges". Just to make it clearer the differences of this section to Section 3.

In the revised manuscript, we renamed the section 4 to "rates of sea level rise at Dublin Port and nearby tide-gauges".

> Can you add a column for the different datums used for each of the datasets, prior to processing? On line 76 "Difficulties in merging the Dublin Port datasets arose from the differing datum definitions" It would be helpful to have Table 1 and Table 2 combined - This is useful to compare the different durations of each record. Also can you check the range of each data record through the paper as the text and table often do not match or are consistent. -Port Authority Annual: duration: 1938-1977. Line 46: " refer to as 1938 to 1988" -Port Authority Monthly: duration 1978-1988: Lines 45- 50. "The Monthly Port Authority data for the period 1987-2001". Is this combined with the first

> dataset refer to in as Port Authority Annual -Harbourmaster:
> duration: 2002-2009. Line 56: "data supplied by the PMSL for
> the period 2001-2009". -NTGN: duration: 2006-2019. Line 60:
> "data for the period 2006-2017"

In the revised manuscript, we added a datumn colomn to Table-1. However, we think combining Table-1 and Table-2 could cause some confusion because the two tables have different purposes. The first one is describing the various data sources used to create a reconciled dataset for Dublin port whereas the second one is introducing the other datasets from other locations that we used to validate the reconciled Dublin port dataset.

We fixed the Table-1 and changed the text accordingly to make sure everything is consistent.

> Figure1: The length of the records shown do not correspond to
> the length of the records in Table.1. For example: Port Author-
> ity (blue)   1968 to 2011?  This may be a plotting problem or
> the resolution of the figure.

Figure 1 is fixed in the revised manuscript and is now consistent with the information given in Table-1.

> Line 89: "we now use our newly merged dataset (..) MSL and
> MHW from 1938 and 2019. As you refer to the period 1938 -
> 2019, why is Figure1 showing the MHW from 1968 to 2019?

Figure 1 is trying to illustrate the consistency among the monthly values of different datasets used for the reconciliation whereas 1938-2019 is the duration of the reconciled dataset with annual resolution. We don't have monthly data pre-1968 and we clarified this in line 84 of the revised manuscript.

> Table 2 and Figure 3: Can you clarify why the time axis is only
> to 2017 (but the records extend to 2019?)

We decided to remove the data post 2016 because of its poor quality. This is clarified in our revised manuscript (Lines 94-97).

---

## Author Response (AR2)

Dear Professor Huthnance,

Many thanks for considering our paper titled 'A newly reconciled data set for identifying sea level rise and variability in Dublin Bay'. We have fixed all the identified typos and badly-worded sentences. We have also address all the other issues identified in the previous round of reviews.

We hope that you now find the paper suitable for publication. If you have any further queries please do not hesitate to get in contact with me.

Yours sincerely,

Amin Shoari Nejad

Line 7. I think CI should be defined as "confidence interval".

In the revised manuscript we are more explicit by stating "credible interval" which is the uncertainty interval calculated from the posterior distribution of a Bayesian model, whereas "confidence interval" is associated with the traditional Frequentist statistical models.

Line 48. You refer to "Annual high and low water . . . for the period 1938 to 2001" but lines 49 to 51 only mention the datum for data up to 1988 and line 54 refers to only "two overlapping years". Why ignore the annual data for 1989-2001?

In line 51 the period "1978-1988" and "two overlapping years" were mistakenly specified. In the revised manuscript the following changes are applied:

"1978-1988" to "from 1978"

"two overlapping" to "the overlapping 15 years (1987–2001)"

Line 74. "The second-high tide". As Reviewer 1 said, on some days there is only one high tide. You should say what procedure allowed for this.

We added the following sentence in line 75 of the revised manuscript to explain the procedure:

"In the case where the time window went into the next calendar day, only a single high tide was recorded for that day."

Table 1. I miss the definition of MTL. Please also give an explicit definition of mean sea level (MSL) so that the difference between MTL and MSL is clear.

Line 49: We added the following line defining MTL:

"Mean tide level (MTL) was calculated by averaging mean high and low waters."

Line 78. Elsewhere (line 62, Table 1) the NTGN dataset only begins in 2007 giving only three years overlap. Please be consistent.

We corrected the line by replacing the "four" with "three".

Line 82. ". . no overlaps . ." but Table 1 shows Port Authority monthly completely covering Harbourmaster. Likewise Figure 1: why does the Port Authority stop at 2001 or 2002 whereas in Table 1 it continues to 2017?

The end date 2017 was mistakenly stated in Table 1. We have corrected it by changing the end date of the Port Authority Monthly dataset from 2017 to 2001.

Table 2. I think this should include the tide gauge type (radar?) as implicitly asked for by Reviewer 1. For the Dublin gauges, I think some of the detail sought by Reviewer is still lacking: "calibration method used to level it in such that low waters were accurate?" "why were high waters less reliable?" "density (and maybe gravity) assumed", "manufacturer?" "absolute or differential sensor?"

Unfortunately we do not have any evidence of a systematic levelling that would make the low waters more accurate but our cross comparison indicates a high level of accuracy between sites. We discuss this in Section 3 of the paper.

Regarding the instruments installed at Arklow, Howth Harbor, Newlyn, and Brest we added the following sentence in line 93:

"The Arklow and Howth Harbour data sets are derived from bubbler gauges. The Newlyn and Brest data are gauges with a long history of use in sea level studies (Bradshaw et al. (2016); Woppelmann et al. (2006))"

We could not find any further information on the manufacturer or other specifications of the tide gauges.

Line 99. ". . malfunction of the Dublin tide gauge . ." (the reader needs to know which tide gauge)

In line 102 of the reviseed manuscript we specified the Dublin Port tide gauge.

Lines 120 and two lines before 145. "mean process" is "statistical mean" or "mean of the process" or simply "mean"?

The expected value of MSL is a stochastic process which is why it is referred to as a mean process. However, we changed the terminology in the revised manuscript to simply "mean" to avoid confusion for the readers of the journal.

Lines 137-138. "Due to the lack of atmospheric data for Dublin Port before 1948, we removed the atmospheric effects after correcting for MSL." This sentence is very unclear. How do you remove atmospheric effects without atmospheric data? What does "correcting for MSL" mean (is it a reference to some procedure in section 3)? Please be explicit.

We clarified our procedure by revising the lines 142-145 as follows:

"The MSL data from Brest is missing between 1944 to 1952, so we decided to limit our SLR rate estimation to 1953-2016 during which the data for all three sites were complete. We first removed the inverse barometer and wind effects on sea level at each site following Frederikse et al. (2017) and Diabate et al. (2021) (we omitted this step in the previous section due to lack of atmospheric data during 1938-1948). "

In Section 3, we explain adjusting MSL using MLW during 1938-2016. Our atmospheric dataset only covers 1948-2016 so if we wanted to account for the atmospheric effects before adjusting MSL then we had to throw away 10 years of data from 1938 to 1948. We examined these 10 years of data and decided that its exclusion would dramatically increase the residual standard deviation in the model. Hence we decided to first adjust the MSL using all data we have and then in Section 4, where we estimate the SLR rates, remove the atmospheric effects for periods post 1952.

Why do figure 6 and Table 5 only start in 1953? Line 137 refers to atmospheric data from 1948.

As explianed above, that is due to lack of MSL data from Brest pre-1953.

---

## Author Response (AR3)

Dear Professor Huthnance,

I would like to thank you for accepting our paper titled 'A newly reconciled data set for identifying sea level rise and variability in Dublin Bay' for publication in Ocean Science.

The technical issue you mentioned is fixed in the revised manuscript and we hope that you now find the paper ready for publication. If you have any further queries please do not hesitate to get in contact with me.

Yours sincerely,

Amin Shoari Nejad